# Néel spin-orbit torque in antiferromagnetic quantum spin and anomalous Hall insulators

Junyu Tang [1], Hantao Zhang[2] & Ran Cheng [1,2,3] ✉

Interplay between topological electrons and magnetic ordering enables efficient electrical control of magnetism. We extend the Kane-Mele model to include the exchange coupling to a collinear antiferromagnetic (AFM) order, which allows the system to exhibit the quantum anomalous Hall and quantum spin Hall effects in the absence of a net magnetization. These topological phases support a staggered Edelstein effect through which an applied electric field can generate opposite non-equilibrium spins on the two AFM sublattices, realizing the Néel-type spin-orbit torque (NSOT). Contrary to known NSOTs in AFM metals driven by conduction currents, our NSOT arises from pure adiabatic currents devoid of Joule heating, while being a bulk effect not carried by the edge currents. By virtue of the NSOT, the electric field of a microwave can drive the AFM resonance with a remarkably high efficiency, outpacing the magnetic field-induced AFM resonance by orders of magnitude in terms of power absorption.

Harnessing topological materials to manipulate magnetism and magnetic dynamics has opened unique opportunities for energy-efficient spintronics. Recently, a new twist in this direction is inspired by the study of intrinsic magnetic topological insulators (iMTIs), in which topological electrons are highly entangled with the magnetic states[1–11]. Thanks to the synergy of electronic and magnetic degrees of freedom, an iMTI could be operated simultaneously as an electric actuator and a magnetic oscillator without the aid of foreign systems[12], forming a mono-structural setup to achieve spintronic functionalities that would otherwise rely on engineered heterostructures.

Meanwhile, because of the insulating nature of iMTIs, the spin-orbit torque (SOT) and the associated internal spin-transfer processes are not driven by Ohm's currents (extrinsic); instead, they arise from the voltage-induced adiabatic currents carried by topological electrons in the valence band (intrinsic)[13–16]. Because adiabatic currents do not incur Joule heating, they can convert 100% of the input electric power into magnetic dynamics[17], leveraging an unprecedented boost of energy efficiency compared to established solutions. While we have recently demonstrated such lossless intrinsic SOT and its ensuing physical effects in a widely recognized iMTI–MnBi$_2$Te$_4$ of a few layer thick[12], the obtained results only revealed the tip of an enormous unexplored iceberg.

In particular, we identify five outstanding questions beyond what could be answered by the iMTI family of materials. First, what non-trivial topological phases are compatible with a given magnetic order, especially the compensated antiferromagnetic (AFM) state? Second, can these topological phases afford the lossless SOT? Third, will the SOT prevail down to the monolayer limit? Fourth, will a strong Néel-type SOT (NSOT), i.e., contrasting fields acting on different sublattices, be supported by symmetry? Finally, what are the dynamical implications of the NSOT in a chosen material? The known case of MnBi$_2$Te$_4$ only provides us with limited information. For instance, the material becomes a compensated AFM (ferromagnetic) system only if there are an even (odd) number of layers, which corresponds to an axion (quantum anomalous Hall) insulator. In either phase, however, the SOT vanishes identically in the monolayer limit while the NSOT is forbidden by symmetry[12,18] within the linear response regime.

To answer these open questions, we investigate a previously overlooked scenario of insulating magnets which could potentially host topological electrons on their own and support intrinsic NSOT. Transition-metal trichalcogenides (TMTs), in their AFM phase, can be effectively described by an extended Kane-Mele model[19,20] including the exchange coupling of electrons and the AFM background. Depending on the spin-orbit coupling (SOC) and a staggered sublattice potential, such a system can exhibit the quantum anomalous Hall (QAH) effect[21,22] and the quantum spin Hall (QSH) effect[23,24] within the same magnetic phase characterized by a fully compensated collinear

[1]Department of Physics and Astronomy, University of California, Riverside, CA, USA. [2]Department of Electrical and Computer Engineering, University of California, Riverside, CA, USA. [3]Department of Materials Science and Engineering, University of California, Riverside, CA, USA. ✉e-mail: rancheng@ucr.edu

AFM order. Moreover, we find that both the QAH and QSH phases support a staggered Edelstein effect, by which an applied electric field can generate opposite non-equilibrium spins acting on different sublattice magnetic moments, realizing the desired intrinsic NSOT without incurring Joule heating as conduction electrons are eliminated.

In stark contrast to the previously reported extrinsic NSOT driven by dissipative charge currents[25–30], our intrinsic NSOT is dissipationless and does not incur Ohm's conduction because only the adiabatic motions of valence electrons are involved. From the perspective of topological materials, our findings distinguish from previous studies by claiming the NSOT, rather than the widely-found SOT, in the topological nontrivial phases. To demonstrate the physical significance, we study the NSOT-induced AFM resonance and benchmark the result against an ordinary AFM resonance. Remarkably, the NSOT renders the electric field of a microwave the dominant driving force, which could overwhelm the direct coupling to the magnetic field, thus enhancing the AFM resonance amplitude by more than one order of magnitude. The high efficiency of this counter-intuitive electric field-driven AFM resonance is quantified by the dynamical susceptibility. Operational-wise, our findings unravel a unique mechanism to leverage the sub-terahertz AFM dynamics devoid of Joule heating by exploiting magnetic topological materials.

## Results and discussion

### Formalism

We extend the Kane-Mele model by including a collinear AFM order which is exchange coupled to the electrons on a 2D honeycomb lattice. As illustrated in Fig. 1, the magnetic moments on the A and B sublattices are oppositely aligned and pointing perpendicular to the plane. The conceived system is characterized by an effective tight-binding Hamiltonian

$$H = t \sum_{\langle i,j \rangle} c_i^\dagger c_j + i\lambda_{soc} \sum_{\langle\langle i,j \rangle\rangle} \nu_{ij} c_i^\dagger s_z c_j + i\lambda_R \sum_{\langle i,j \rangle} c_i^\dagger (\mathbf{s} \times \hat{\mathbf{d}}_{ij})_z c_j$$
$$+ \lambda_v \sum_i c_i^\dagger \xi_i c_i + \lambda_{ex} \sum_i c_i^\dagger (\mathbf{m}_i \cdot \mathbf{s}) c_i, \tag{1}$$

where $c_i^\dagger (c_i)$ is the electron creation (annihilation) operator on site $i$, with the spin index omitted for succinctness. In Eq. (1), the first term represents the nearest neighbor hopping. The second term is the intrinsic SOC which affects the next-nearest neighbor hopping, where $\nu_{ij} = 2/\sqrt{3}(\hat{\mathbf{d}}_1 \times \hat{\mathbf{d}}_2)_z = \pm 1$ with $\hat{\mathbf{d}}_1$ and $\hat{\mathbf{d}}_2$ being the two unit vectors along the 120° bonds connecting $i$ and $j$. The third term is the Rashba SOC arising from the broken mirror symmetry (with $z$ normal), where $\hat{\mathbf{d}}_{ij}$ is the unit vector connecting the nearest-neighboring sites $i$ and $j$, and $\mathbf{s}$ is the vector of Pauli matrices for the spin degree of freedom. The fourth term is the staggered potential where $\xi_i = \pm 1$ flips sign on the A and B sublattices as shown in the Fig. 1, breaking the $C_2$ symmetry about the $x$ axis. The last term represents the exchange coupling between the electrons and the local magnetic moments, where $\mathbf{m}_i = \pm\hat{\mathbf{z}}$ is the unit magnetic vector on site $i$.

### Topological phases

If the exchange coupling $\lambda_{ex}$ vanishes, the Hamiltonian preserves the time-reversal symmetry. Then for a sufficiently large $\lambda_{soc}$, the system can exhibit the QSH phase characterized by the $Z_2$ number[19]. Now, with a finite $\lambda_{ex}$, the time-reversal symmetry is broken so the $Z_2$ number becomes ill-defined. Moreover, the QSH phase has a vanishing Chern number ($C = 0$), so it could only be distinguished from the normal insulator (NI) phase by the spin Chern number $C_s = (C_\uparrow - C_\downarrow)/2$[31,32].

We first draw the phase diagrams of the Chern number $C$ in Fig. 2a, b by navigating $\lambda_{soc}$, $\lambda_v$ and $\lambda_R$. To ensure a proper quantization of the topological invariant, we impose an upper limit of $0.1t$ for the values of $\lambda_{soc}$, $\lambda_R$, $\lambda_{ex}$ and $\lambda_v$ so as to maintain a global band gap. We find three distinct phases on these two diagrams. The QSH state only appears at

large $\lambda_{soc}$. At $\lambda_v = 0$, the threshold of QSH is about $\lambda_{soc} = \pm 0.03t$. Two observations are in order. First, different from the QSH state, the QAH state requires a nonzero staggered potential $\lambda_v$. This is because the staggered potential breaks the $\mathcal{PT}$ symmetry (combined inversion and time reversal), enabling a non-zero Berry curvature[33–35]. Second, while the Chern number flips sign when either $\lambda_v$ or $\lambda_{soc}$ flips sign, it remains the same regardless of $\lambda_R$, because $+\lambda_R$ and $-\lambda_R$ are related by a mirror reflection $z \to -z$, which does not change the sign of $C$. In Fig. S1, we also provide the phase diagrams for other combinations of parameters (e.g., $\lambda_{soc}$ and $\lambda_{ex}$). We conclude that the sign of the total Chern number is determined by $\text{sign}[C] = \text{sign}[\lambda_{soc}]\text{sign}[\lambda_v]\text{sign}[\lambda_{ex}]$.

We next plot the phase diagrams of the spin Chern number $C_s$ in Fig. 2c, d, corresponding to the results in Fig. 2a, b, respectively. As expected, $C_s$ in the QSH state is quantized to be $\pm 1$. In the QAH state, however, $C_s$ is quantized to be $\pm 0.5$, which indicates that the chiral edge electrons only carry one spin species (see Fig. S2 for further details). It is important to note that the spin Chern number near the phase boundaries is not exactly quantized or half quantized, because the spin is not a strictly conserved quantity in the presence of a finite Rashba SOC. Concerning the sign flip of $C_s$, we observe a quite different pattern as compared to $C$. For example, $C_s$ is even in $\lambda_v$ while being odd in $\lambda_{soc}$. This can be understood from definition of spin currents: if the spin polarization and the flowing direction both flip sign, a spin current will remain unchanged.

To further confirm the system topology revealed by the phase diagrams, we plot in Fig. 2e, f the band structures of our Hamiltonian truncated in the $y$ direction with $N = 40$ unit cells (i.e., a nanoribbon periodic only in the $x$ direction). For $\lambda_{soc} = 0.02t$, the system is in the QAH state, where only one pair of chiral edge states with the same spin polarization but opposite group velocities emerges in the band gap. For $\lambda_{soc} = 0.05t$, the system transitions into the QSH phase, where two pairs of chiral edge states appear in the bulk gap with opposite spin polarizations.

### Néel-type spin-orbit torque

Having obtained the band topology with broken time-reversal symmetry introduced by the AFM order, we are in a good position to explore the interplay between electron transport and magnetic dynamics. For insulating systems where the ordinary spin Hall effect is suppressed, applying an (in-plane) electric field $\mathbf{E}$ can directly generate non-equilibrium spin accumulation through the Edelstein effect[36]. The induced spin accumulation can in turn excite magnetic dynamics through the SOT[37–40]. In our context, it is important to discern different AFM sublattices in the non-equilibrium spin generation. While the average component $\delta\mathbf{S} = (\delta\mathbf{S}^A + \delta\mathbf{S}^B)/2$ (due to the Edelstein effect) leads to the ordinary SOT, the contrasting component $\delta\mathbf{N} = (\delta\mathbf{S}^A - \delta\mathbf{S}^B)/2$ (due to the staggered Edelstein effect) leads to the NSOT[25]. As we consider insulating magnets where the Fermi energy $\varepsilon_F$ lies in the bulk gap, $\delta\mathbf{S}$ and $\delta\mathbf{N}$ only involve the Fermi-sea contribution. Within the linear response regime, we can express the contrasting spin accumulation as[25,39,41]

$$\delta\mathbf{N} = \frac{e\hbar^2}{2} \sum_{\epsilon_n < \varepsilon_F < \epsilon_m} \int_{BZ} \frac{dk^2}{(2\pi)^2} \text{Im}\left[\langle n|\mathbf{s} \otimes \tau_3|m\rangle\langle m|\mathbf{v} \cdot \mathbf{E}|n\rangle\right] \frac{(\epsilon_n - \epsilon_m)^2 - \Gamma^2}{[(\epsilon_n - \epsilon_m)^2 + \Gamma^2]^2}, \tag{2}$$

where $\mathbf{v}$ is the velocity operator and $\Gamma$ is the energy broadening due to disorder. The average spin accumulation $\delta\mathbf{S}$ follows a similar formula with the pseudo-spin Pauli matrix (acting on the sublattices) $\tau_3$ replaced by the identity matrix. Unlike the Fermi-level contribution, here $\delta\mathbf{N}$ does not diverge even in the clean limit $\Gamma \to 0$ where Eq. (2) reduces to a formula similar to the spin Chern number [see Eq. (8)]. In the following, we will take representative values for the exchange interaction $\lambda_{ex} = 0.1t = 100$ meV and for the band broadening $\Gamma = 20$ meV.

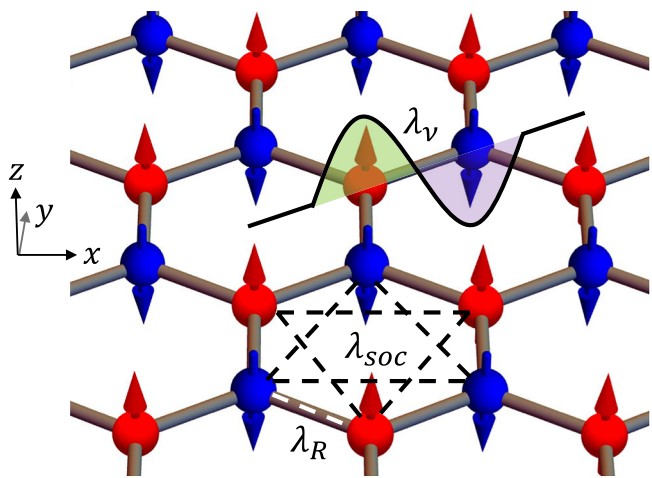

**Fig. 1 | Schematic of a honeycomb lattice with G-type AFM order.** The wavy potential well represents the staggered potential on A and B sublattices. The intrinsic SOC and Rashba SOC introduce extra phases in the nearest neighbor (white dashed line) and next nearest neighbor hopping (black dashed line), respectively. The coordinates axis are shown on the left.

Without sacrificing generality, we set the **E** field in the $x$ direction and calculate the non-equilibrium spin accumulation on each sublattice: $\delta \mathbf{S}^A = (\delta \mathbf{S} + \delta \mathbf{N})/2$ and $\delta \mathbf{S}^B = (\delta \mathbf{S} - \delta \mathbf{N})/2$. Figure 3a, b plot $\delta S_x^A$ and $\delta S_x^B$ as functions of $\lambda_{soc}$ and $\lambda_R$, while the $y$ and $z$ components are found to be zero. Remarkably, we find that $\delta S_x^A$ and $\delta S_x^B$ are exactly opposite to each other so long as $\lambda_\nu = 0$, meaning that only the NSOT exists whereas the SOT vanishes (i.e., $\delta \mathbf{S} = 0$). As a consistency check, the way $\delta \mathbf{S}^{A(B)}$ varies over the direction of **E** is shown in Fig. S4, where the contrasting feature $\delta S_{x(y)}^A = - \delta S_{x(y)}^B$ persists. Figure 3c, d schematically show the difference between the SOT and NSOT, driven by $\delta \mathbf{S}^A = \delta \mathbf{S}^B$ and $\delta \mathbf{S}^A = -\delta \mathbf{S}^B$, respectively. In the clean limit $\Gamma \to 0$, the spin accumulation on each sublattice is directly related to the Berry curvature residing in the mixed space of crystal momentum and magnetization[13,17]:

$$\delta S_\nu^{A(B)} = \frac{e\hbar}{2\lambda_{ex}} E_\mu \left\langle \Omega_{\mu\nu}^{km^{A(B)}} \right\rangle, \quad (3)$$

where $\langle \cdots \rangle = V_{uc}/(2\pi)^2 \sum_n \int d^2k f[\varepsilon_n(\mathbf{k})] (\cdots)$ denotes the average over the first Brillouin zone with $V_{uc}$ the unit cell volume (area) and the $f$ the Fermi distribution. For $\Gamma \neq 0$, the Berry curvature is dressed by the broadening $\Gamma$, becoming the integrand of Eq. (2). For $\lambda_\nu = 0$, the Berry curvature assumes a staggered pattern $\Omega_{\mu\nu}^{km^A} = - \Omega_{\mu\nu}^{km^B}$, hence $\delta \mathbf{S}^A = -\delta \mathbf{S}^B$. A non-zero $\lambda_\nu$ would render $|\delta S_x^A| \neq |\delta S_x^B|$ owing to the introduction of staggered potentials on the sublattices (see Fig. S3), which leads to a finite $\delta \mathbf{S}$ on top of the $\delta \mathbf{N}$, hence inducing a nonzero SOT besides the NSOT.

In the phase diagram, the dashed lines mark where the global gap reduces to 1 meV. For a large $|\lambda_R|$ beyond these boundaries, the global gap will be smaller than 1 meV, making it difficult to restrict $\varepsilon_F$ in the gap and, more seriously, less practical to guarantee the adiabatic condition (to be clear in the next section). Therefore, we should focus on the central region of Fig. 3a, b enclosed by the dashed lines to safely ignore the Fermi-surface contribution to $\delta \mathbf{S}^{A(B)}$.

The NSOT is known for being able to switch the Néel order $\mathbf{n} = (\mathbf{m}^A - \mathbf{m}^B)/2$ in non-centrosymmetric AFM metals[25–30]. But previous experimental studies are limited to current-induced NSOT. By contrast, our predicted NSOT is in principle free of dissipation because it is mediated by the adiabatic motions of valence electrons, incurring no Ohm's current as no conduction electrons are involved in the generation of spin torques. The adiabatic origin of the NSOT, similar to the SOT previously claimed in iMTIs, is also reflected by its Berry-curvature

origin discussed above. We emphasize that the Berry curvature $\Omega^{km}$ relevant to the NSOT is physically distinct from the momentum-space Berry curvature $\Omega^{kk}$ that determines the band topology[12,16].

To better clarify this subtle point, we plot in Fig. 4a the sublattice spin accumulations $\delta \mathbf{S}^A$ and $\delta \mathbf{S}^B$ as functions of $\lambda_{soc}$ with vanishing $\lambda_\nu = 0$, i.e., a vertical cut at $\lambda_R = 0.02t$ in Fig. 3. As a comparison, we also plot the results with a non-vanishing staggered potential $\lambda_\nu = 0.04t$ in Fig. 4b. The NI, QAH and QSH regions are shaded in light green, purple and orange, respectively. Three key observations are in order. First, although $\delta \mathbf{S}^{A,B}$ turn out to be slightly larger in the QAH and QSH states, they remain finite even in the topologically trivial phase, suggesting that the NSOT cannot be fully characterized by the band topology. Second, $\delta \mathbf{S}^{A,B}$ exhibit sudden jumps at the QAH-NI and QSH-NI transitions. These jumps would formally diverge in the clean limit $\Gamma \to 0$ due to gap closing; but in our plots, a finite $\Gamma = 20$ meV is added to suppress the divergence. Third, the hallmark NSOT signature, $\mathrm{sgn}(\delta S^A) = - \mathrm{sgn}(\delta S^B)$, is most robust in the QSH phase. While a finite $\lambda_\nu$ enables a QAH phase interpolating the QSH and NI phases in Fig. 4b, it also imbalances the potentials on the two sublattices, rendering $\delta \mathbf{S}^A$ and $\delta \mathbf{S}^B$ different in magnitude. Specifically, we have $\delta S_x^A/\delta S_x^B \approx -1.28$ at $\lambda_{soc} = 0.05t$, whereas in Fig. 4a with $\lambda_\nu = 0$ we have exactly $\delta S_x^A/\delta S_x^B = -1$ for all $\lambda_{soc}$. The dependencies of $\delta \mathbf{S}^{A,B}$ on $\lambda_R$ and $\lambda_{ex}$ are shown in Fig. S3.

## Electric field-driven antiferromagnetic resonance

To demonstrate the dynamical consequences of the predicted NSOT, we now study the AFM resonance driven by an AC electric field. In terms of the unit vectors of the sublattice magnetic moments, the governing Landau-Lifshitz-Gilbert (LLG) equations are:

$$\dot{\mathbf{m}}^A = \gamma \left[ -\mathcal{H}_J \mathbf{m}^B + \mathcal{H}_\parallel (\mathbf{e}_\parallel \cdot \mathbf{m}^A) \mathbf{e}_\parallel + \mathbf{H}_0 + \mathbf{h}_D^A \right] \times \mathbf{m}^A + \alpha_0 \mathbf{m}^A \times \dot{\mathbf{m}}^A \quad (4a)$$

$$\dot{\mathbf{m}}^B = \gamma \left[ -\mathcal{H}_J \mathbf{m}^A + \mathcal{H}_\parallel (\mathbf{e}_\parallel \cdot \mathbf{m}^B) \mathbf{e}_\parallel + \mathbf{H}_0 + \mathbf{h}_D^B \right] \times \mathbf{m}^B + \alpha_0 \mathbf{m}^B \times \dot{\mathbf{m}}^B, \quad (4b)$$

where $\gamma > 0$ is the gyromagnetic ratio, $\mathcal{H}_J$ is the AFM exchange field (summed over all nearest neighbors), $\mathcal{H}_\parallel$ is the anisotropy field for the easy axis $\mathbf{e}_\parallel$, $\mathbf{H}_0$ is the external static field, and $\alpha_0$ is the Gilbert damping constant. For simplicity, let $\mathbf{e}_\parallel$ be the $z$ axis and $\mathbf{H}_0$ be applied along $\mathbf{e}_\parallel$, lifting the degeneracy of the AFM resonance modes.

Under a microwave irradiation, the oscillating driving field $\mathbf{h}_D^{A(B)}$ can arise either directly from the magnetic field $\mathbf{h}_{rf}$ or indirectly from the NSOT field

$$\mathbf{h}_{NS}^{A(B)} = -\frac{2\lambda_{ex}}{\hbar m_s} \delta \mathbf{S}^{A(B)} \quad (5)$$

produced by the electric field $\mathbf{E}_{rf}$, where $m_s$ is the sublattice magnetic moment. According to Eq. (2), $\delta \mathbf{S}^{A/B} = (\delta \mathbf{S} \pm \delta \mathbf{N})/2$ decreases monotonically with an increasing $\lambda_{ex}$ because the topological band gap in our model is primarily determined by $\lambda_{ex}$. Consequently, the NSOT field determined by Eq. (5), with a linear dependence on $\lambda_{ex}$ in its front factor, remains insensitive to the change of $\lambda_{ex}$. Of the two mechanisms, $\mathbf{h}_{rf}$ ($\mathbf{h}_{NS}$) is perpendicular (parallel) to $\mathbf{E}_{rf}$ and is the same (opposite) on each sublattice. Based on Maxwell's equations, a microwave with $|\mathbf{E}_{rf}| = 1 \mathrm{V}/\mu m$ has a magnetic field $|\mathbf{h}_{rf}| = 33$ Gauss. The same electric field can generate a maximum non-equilibrium spin of $0.85 \times 10^{-5} \hbar$ per sublattice according to Fig. 3, which converts to an effective NSOT field $|\mathbf{h}_{NS}| = 59$ Gauss for $m_s = 5\mu_B$. While we are not able to locate a specific material on the phase diagram Fig. 3, it is instructive to chose a point where $|\mathbf{h}_{rf}| = |\mathbf{h}_{NS}|$, so we can determine how their distinct symmetry (uniform $\mathbf{h}_{rf}$ versus opposite $\mathbf{h}_{NS}$ on the two sublattices) could lead to dramatically different microwave absorptions with the onset of AFM resonance.

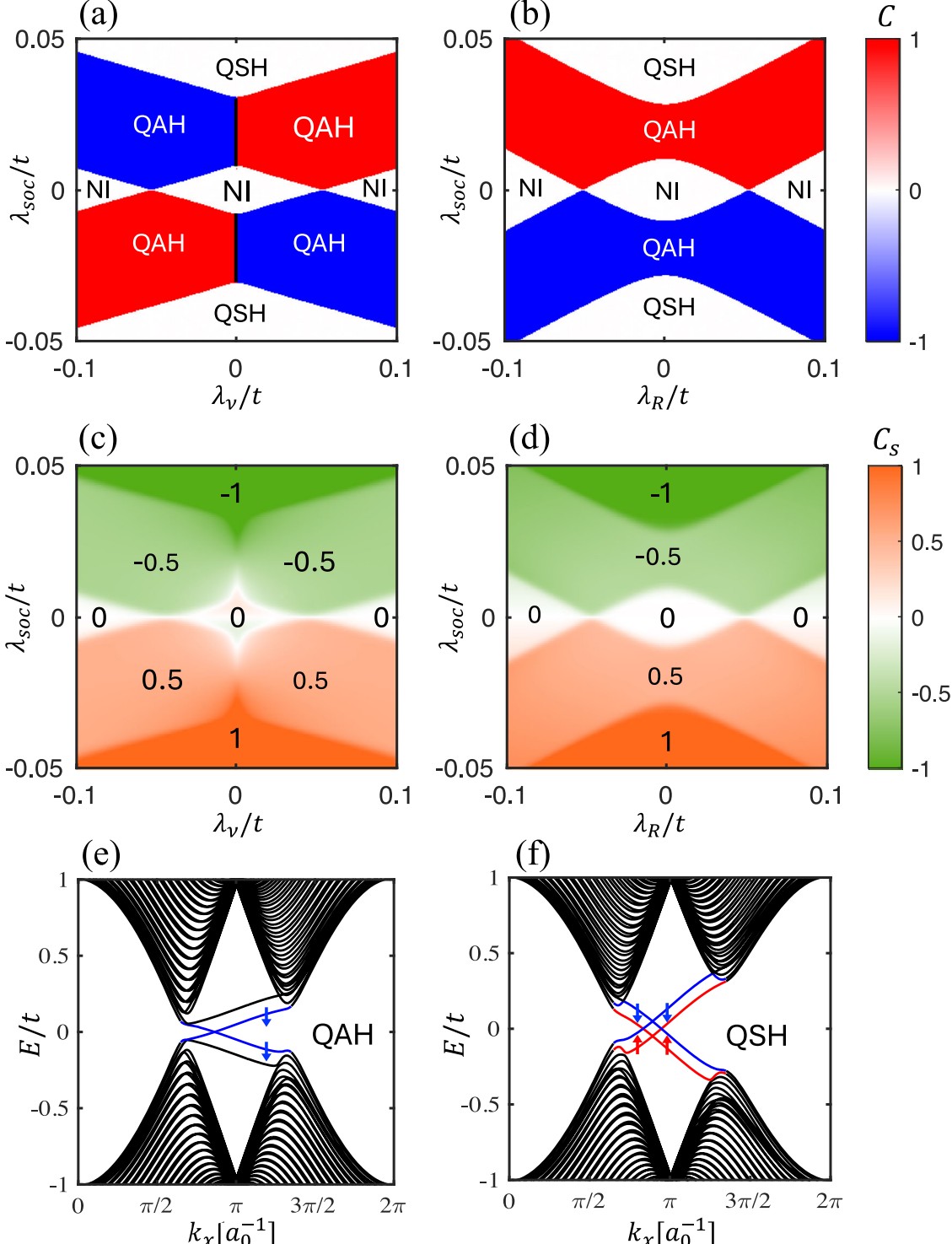

**Fig. 2 | Electronic phase diagrams and edge states.** Chern number (**a**, **b**) and spin Chern number (**c**, **d**) with respect to intrinsic SOC strength $\lambda_{soc}$ for different staggered potential and Rashba SOC strength $\lambda_R$. In **a**, **c**, $\lambda_R$ is fixed to be $0.05t$. In **b**, **d**, $\lambda_v$ is fixed to be $0.05t$. Band structure of a finite system, with 40 unit cells in the $y$ direction, for **e** $\lambda_{soc} = 0.02t$ and **f** $\lambda_{soc} = 0.05t$. In both **e**, **f**, $\lambda_R = 0.025t$ and $\lambda_v = 0.05t$. The edge states in the bulk band gap are colored blue or red depending on the spin polarization. $a_0$ is the lattice constant. In all cases, $\lambda_{ex} = 0.1t$.

To this end, we focus on the point at $\lambda_{soc} = 0.05t$, $\lambda_R = 0.072t$, and $\lambda_v = 0$, which lies in the QSH phase. Here, an electric field of $0.5V/\mu m$ will produce a staggered spin accumulation $\delta S_x^A = -\delta S_x^B = -2.4 \times 10^{-6}\hbar$ per unit cell (about 64% of the maximum capacity on the phase diagram), which corresponds to $h_{NS} = 16.5\,G$, matching the real magnetic field $h_{rf}$ of the same electromagnetic wave. We then consider a linearly polarized microwave incident from the $y$ direction with either

$h_{rf}$ or $E_{rf}$ (hence the $h_{NS}$), but not both, parallel to the $x$ axis, as illustrated in Fig. 5a, b, respectively. Such experimental conditions can be typically realized with the Voigt geometry rather than the Faraday geometry[42]. Under the device geometry in Fig. 5a [Fig. 5b], the electric field $E_{rf}$ (magnetic field $h_{rf}$) is collinear with the magnetic moments so that only $h_{rf}$ ($E_{rf}$) drives the AFM dynamics, separating the NSOT-induced resonance from the ordinary AFM resonance.

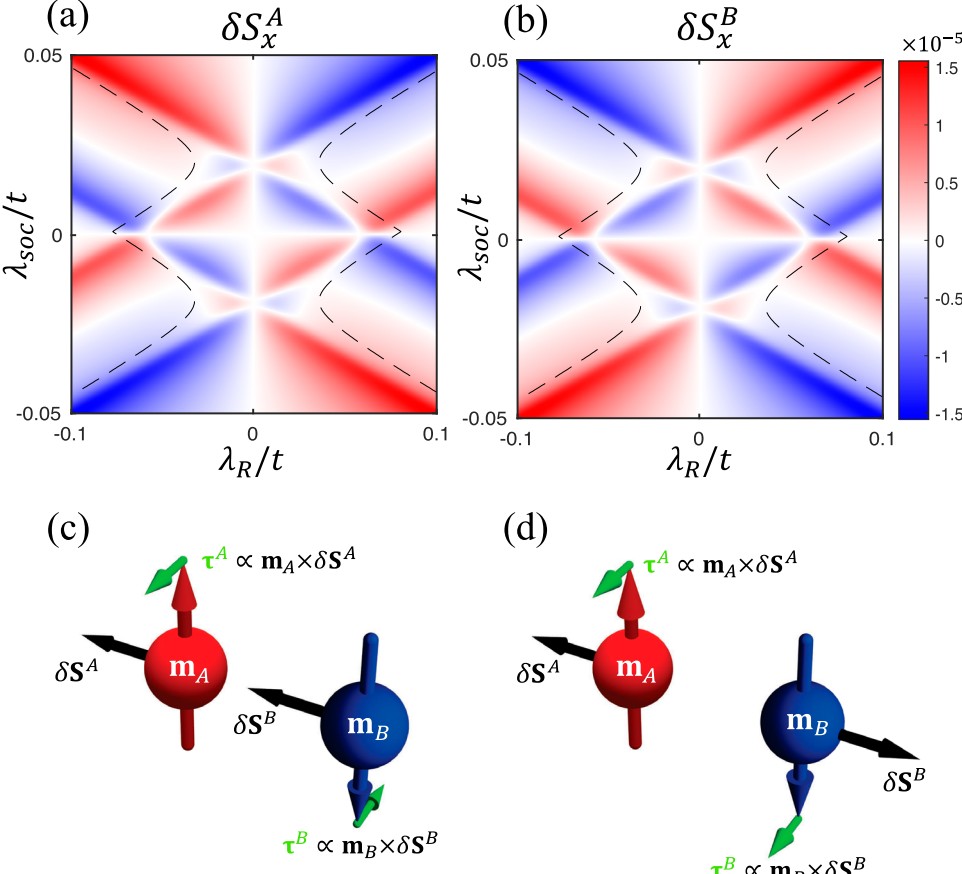

**Fig. 3 | Non-equilibrium spin accumulations and their ensuing torques.**
**a**, **b** Phase diagrams of $\delta \mathbf{S}$ per unit cell (in units of $\hbar/2$) for each sublattice induced by $E_x = 1$V/µm for $\lambda_\nu = 0$. The dashed lines mark where the global band gap is reduced to 1 meV with an increasing $|\lambda_R|$. Illustrative comparison between: **c** ordinary SOT induced by a uniform spin accumulation $\delta \mathbf{S}^A = \delta \mathbf{S}^B$, and **d** NSOT induced by a contrasting spin accumulation $\delta \mathbf{S}^A = -\delta \mathbf{S}^B$.

Next, we study the time evolution of the Néel vector $\mathbf{n}(t)$ by numerically solving the LLG equations (4), where parameters are given typical values in 2D honeycomb TMTs (such as MnPS$_3$ and its variances[43,44]): $m_s = 5\mu_B$, $\mathcal{H}_J = 35$ T and $\mathcal{H}_\parallel = 0.16$ T. Figure 5a, b plot the transverse components $n_x(t)$ and $n_y(t)$ for the two distinct cases under the resonance condition of the low-frequency mode: $f = \gamma\sqrt{\mathcal{H}_\parallel(\mathcal{H}_\parallel + 2\mathcal{H}_J)} - \gamma H_0$ [45], where $H_0 = 1.5$ T is applied along the $+z$ direction, yielding the low-frequency mode left-handed as illustrated by the inset of Fig. 5d. With our chosen parameters, this $H_0$ field is well below the spin-flop threshold while separating the left-handed and right-handed modes by 84 GHz. We emphasize that the vertical axis in Fig. 5b has a scale 15 times larger than that in Fig. 5a, and the amplitude of AFM resonance is about 20 times larger in Fig. 5b where the magnetic dynamics is activated by the $\mathbf{E}_{rf}$ field (through the NSOT). To further confirm this point, we plot in Fig. S5 the case of a vertically incident microwave (i.e., the Faraday geometry) where both $\mathbf{h}_{rf}$ and $\mathbf{E}_{rf}$ can drive the magnetic dynamics. The result is hardly distinguishable from Fig. 5b, indicating that the NSOT overwhelms the Zeeman coupling in driving the AFM resonance.

If the driving frequency $f$ (in energy scale $2\pi\hbar f$) is comparable with the band gap[18,46], Eq. (2) as a Berry phase result will become invalid because the adiabatic condition is broken and the transitions from the valence band to the conduction band become substantial. But the typical AFM resonance frequency we are considering is at most in the sub-terahertz range, where $2\pi\hbar f$~0.2 meV is much smaller than the band gap so long as we stay fairly far away from the phase transition point.

Were not the NSOT generation, the electric field $\mathbf{E}_{rf}$ is not even able to drive the spin dynamics, let alone entailing an enhanced resonance amplitude. To quantify the resonance absorption of the microwave, we linearize the LLG equations (4) using the vectorial phasor representation: $\mathbf{m}^{A(B)} = \text{Re}[\tilde{\mathbf{m}}^{A(B)}e^{i\omega t}]$ and $\mathbf{h}^{A(B)} = \text{Re}[\tilde{\mathbf{h}}_D^{A(B)}e^{i\omega t}]$, with either $\tilde{\mathbf{h}}_D^A = \tilde{\mathbf{h}}_D^B = \tilde{h}_{rf}\hat{\mathbf{x}}$ or $\tilde{\mathbf{h}}_D^A = -\tilde{\mathbf{h}}_D^B = \tilde{h}_{NS}\hat{\mathbf{x}}$ depending on which component acts as the driving field. Since we have fixed the driving field to be polarized along $\mathbf{x}$, the dynamical susceptibility tensor of the Néel vector reduces to a vector $\tilde{\boldsymbol{\chi}}_\perp^n(\omega) = \{\tilde{\chi}_x^n(\omega), \tilde{\chi}_y^n(\omega)\}$ defined by

$$\tilde{n}_{x(y)} = \tilde{\chi}_{x(y)}^n(\omega)\gamma\tilde{h}_D, \tag{6}$$

where $\tilde{h}_D = \tilde{h}_{rf}$ for the geometry in Fig. 5a, whereas $\tilde{h}_D = \tilde{h}_{NS}$ for the geometry in Fig. 5b. For simplicity, we set the initial phase of $\tilde{h}_D$ zero, so the phase difference between $\tilde{n}$ and $\tilde{h}_D$ is embedded in the phase of $\tilde{\chi}^n(\omega)$. We numerically plot the amplitude $|\tilde{\boldsymbol{\chi}}_\perp^n| \equiv \sqrt{|\tilde{\chi}_x^n|^2 + |\tilde{\chi}_y^n|^2}$ and the phase $\text{Arg}[\tilde{\boldsymbol{\chi}}_\perp^n] \equiv \text{Arg}[\tilde{\chi}_x^n] - \text{Arg}[\tilde{\chi}_y^n]$ (in different colors) as a function of the frequency for the $\mathbf{h}_{rf}$-driven resonance and the $\mathbf{E}_{rf}$-driven resonance in Fig. 5c, d, respectively. Similar to the time-domain plots, here we intentionally adopt very different scales for the ordinates in Fig. 5c, d, which clearly shows that $|\tilde{\boldsymbol{\chi}}_\perp^n|$ (hence the microwave absorption) is about 20 times larger when $\mathbf{E}_{rf}$ activates the resonance (via the NSOT), as compared with the ordinary $\mathbf{h}_{rf}$-driven mechanism (via direct Zeeman coupling). Basing on $\text{Arg}[\tilde{\boldsymbol{\chi}}_\perp^n]$, we can further tell that the low-frequency mode indeed exhibits a left-handed precession of the Néel vector while the high-frequency mode is right-handed.

For ferromagnetic resonances[47], the power absorption rate at the resonance point is simply proportional to the amplitude of the

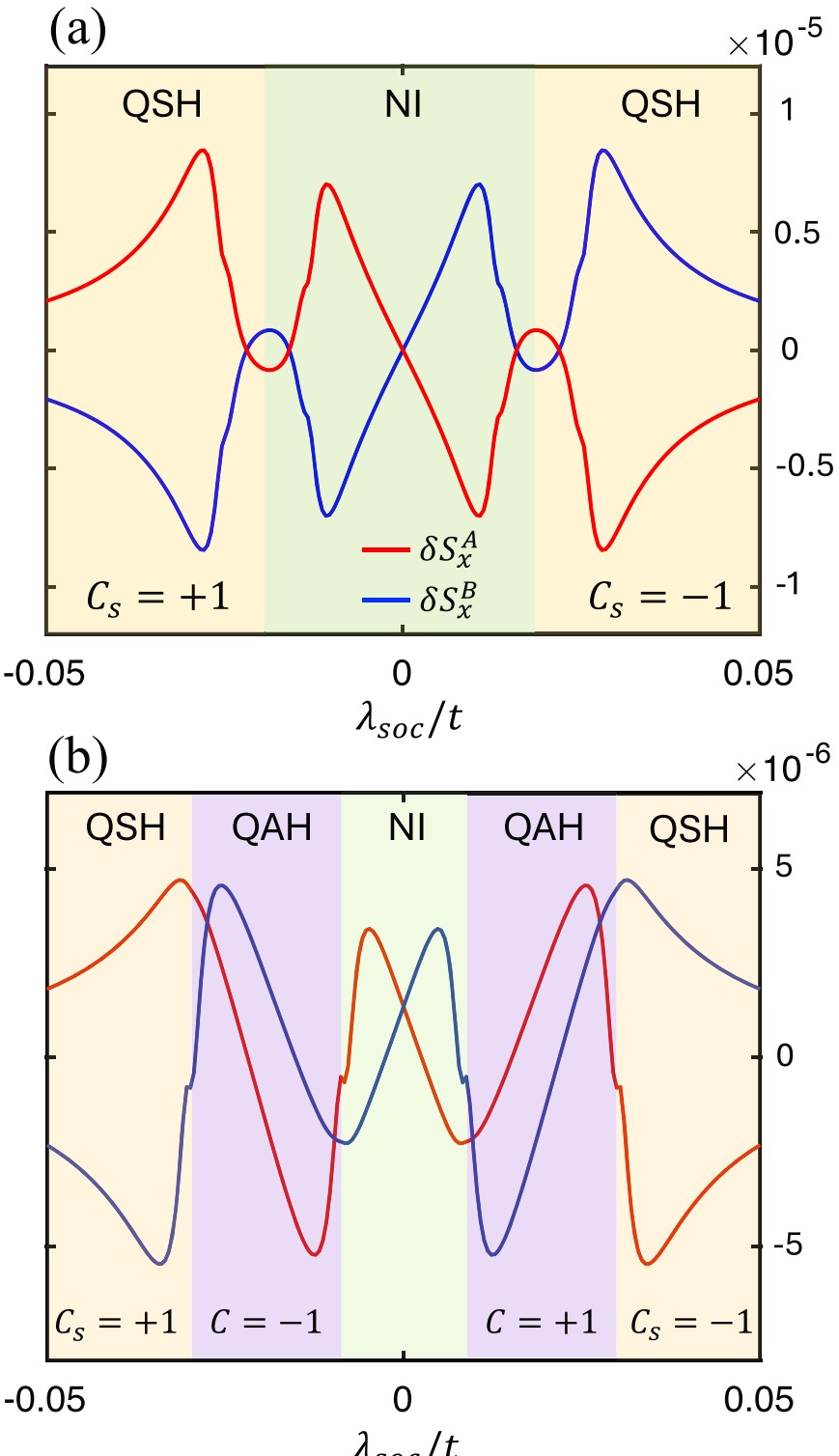

**Fig. 4 | Non-equilibrium spin accumulations across different phases.** $\delta\mathbf{S}$ per unit cell (in units of $\hbar/2$) for each sublattice as a function of $\lambda_{soc}$ with **a** $\lambda_v = 0$ and **b** $\lambda_v = 0.04t$. For each topological nontrivial regions, the corresponding Chern number or spin Chern number are labeled in the bottom. In both figures, we adopt $\lambda_R = 0.02t$, $\lambda_{ex} = 0.1t = 100$ meV, $E_x = 1$V/μm.

dynamical susceptibility, given a fixed strength of the driving field. However, the case is subtly different when we turn to AFM resonances and look into the dynamical susceptibility of the Néel vector $\tilde{\chi}_\perp^n(\omega)$. Even though we have considered a particular case where $h_{rf} = h_{NS}$, the actual power absorption rate under the $\mathbf{E}_{rf}$-driven mechanism is not naïvely proportional to $|\tilde{\chi}_\perp^n|$ ascribing to the staggered nature of the

NSOT field ($\mathbf{h}_{NS}^A = -\mathbf{h}_{NS}^B$). In "Method" (Sec. B), we rigorously derive the time-averaged dissipation power for each mechanism, which allows us to quantify the ratio of the microwave power absorption rate under the two mechanisms as $\bar{P}_E/\bar{P}_H \approx 438.5$. The significantly enhanced microwave power absorption rate will greatly facilitate the detection of spin-torque excited AFM resonance.

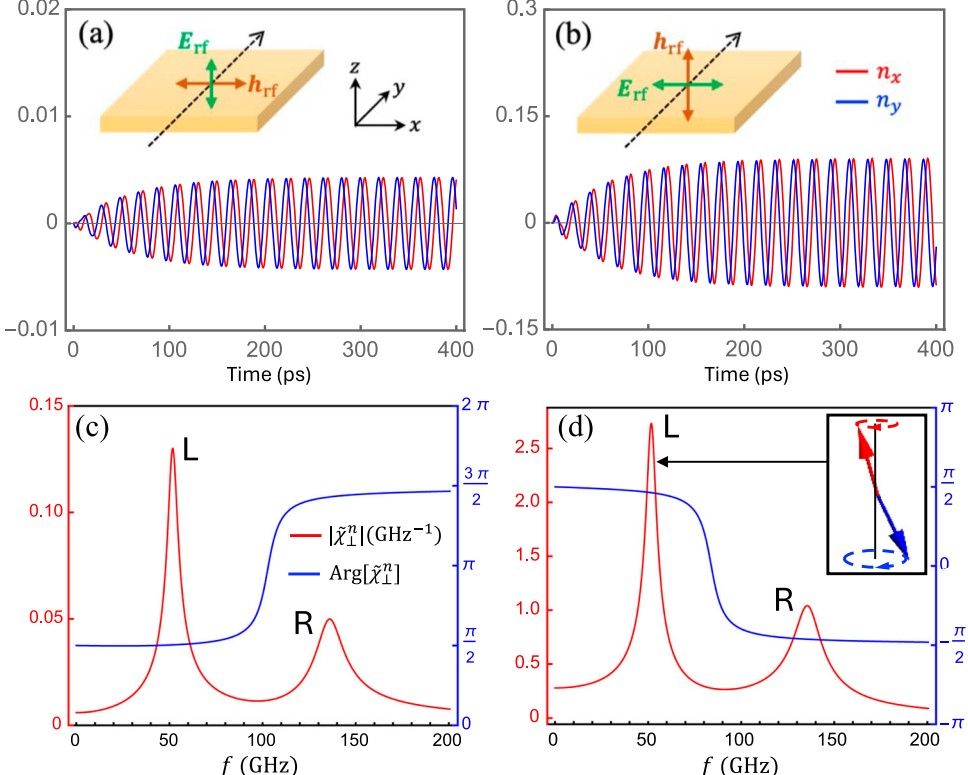

**Fig. 5 | Antiferromagnetic resonances.** Time evolution of the Néel order $\mathbf{n}(t)$ at the resonance of the left-handed mode (with $f = 51.9$ GHz set by a bias field $H_0 = 1.5$ T) for **a** $\mathbf{h}_{\text{rf}}$-driven configuration, and **b** $\mathbf{E}_{\text{rf}}$-driven configuration. **c**, **d** are the corresponding amplitude (red, left axis) and phase (blue, right axis) of the dynamical susceptibility $\tilde{\chi}_\perp^n$ as a function of driving frequency $f$, where the low-frequency mode is left-handed (inset). Parameters: $H_J = 35$ T, $H_\parallel = 0.16$ T, $\alpha_0 = 0.005$, and $E_{\text{rf}} = 0.5$ V/μm (corresponding to $h_{\text{rf}} = 16.5$ Gs).

## Final remarks

To intuitively understand the pronounced difference in microwave absorption between the two mechanisms, we resort to the symmetry of NSOT. In contrast to the uniform Zeeman field $\mathbf{h}_{\text{rf}}$ tending to kick $\mathbf{m}^A$ and $\mathbf{m}^B$ towards opposite directions, the NSOT field is itself opposite on the two sublattices, $\mathbf{h}_{\text{NS}}^A = -\mathbf{h}_{\text{NS}}^B$ [see Fig. 3d], which drives the two magnetic moments towards the same direction, thus amplifying their non-collinearity. Consequently, the strong exchange interaction between $\mathbf{m}^A$ and $\mathbf{m}^B$ is leveraged to enhance the efficiency of magnetic dynamics, resulting in a much stronger absorption of the microwave.

In summary, we have studied the exotic topological phases, and the spin-torque generations in these phases, based on a G-type AFM material with a honeycomb lattice in the presence of the intrinsic SOC, the Rashba SOC and a staggered potential. We find that the highly non-trivial Néel-type SOT can not only be induced by an applied electrical field without producing Joule heating but also be utilized to drive the AFM resonance at a remarkably high efficiency, which, even under a conservative estimate, is more than one order of magnitude larger than the traditional AFM resonance relying on the Zeeman coupling. Our significant findings open an exciting way for exploiting the unique spintronic properties of AFM topological phases to achieve sub-terahertz AFM magnetic dynamics.

## Methods

### Chern number and spin Chern number

The topological Chern number is calculated by the Kubo formula[48,49]:

$$C = -2\hbar^2 \sum_{\epsilon_n < \epsilon_f < \epsilon_m} \text{Im} \int_{\text{BZ}} \frac{dk^2}{2\pi} \frac{\langle n|\hat{v}_x|m\rangle\langle m|\hat{v}_y|n\rangle}{(\epsilon_n - \epsilon_m)^2}, \quad (7)$$

where $\hbar$ is the reduced Planck constant, $\hat{v}_i = \partial\hat{H}/\hbar\partial k_i$ is the velocity operator and $|n\rangle$ is eigenstate corresponding to $\epsilon_n$. Similarly, the spin Chern number $C_s = (C_\uparrow - C_\downarrow)/2$ is[31,32]:

$$C_s = -2\hbar \sum_{\epsilon_n < \epsilon_f < \epsilon_m} \text{Im} \int_{\text{BZ}} \frac{dk^2}{2\pi} \frac{\langle n|\hat{j}_x^s|m\rangle\langle m|\hat{v}_y|n\rangle}{(\epsilon_n - \epsilon_m)^2}, \quad (8)$$

where $\hat{j}_x^s = \hbar(s_z\hat{v}_x + \hat{v}_x s_z)/4$ is the spin-current operator along $x$ with the spin polarization along $z$. The numerical calculation of the band structure and the spatial distribution of the scattering states are performed by Kwant[50].

### Microwave power absorption

When a steady-state dynamics is established, the density of microwave power absorption $P_H$ should be fully balanced by the magnetic dissipation power density

$$P_m = -\frac{m_s}{V_{\text{uc}}}\mathbf{h}_D \cdot (\dot{\mathbf{m}}^A + \dot{\mathbf{m}}^B) \quad (9)$$

upon time averaging, where $V_{\text{uc}}$ is the unit cell volume (area).

Case I. When the driving field $\mathbf{h}_D$ is the magnetic component $\mathbf{h}_{\text{rf}}$ of the electromagnetic wave, which is polarized along $x$ [Fig. 5a], our convention assumes $h_{\text{rf}}(t) = \text{Re}[\tilde{h}_{\text{rf}}e^{i\omega t}] = H_x\cos(\omega t)$, and $\mathbf{m}^{A(B)}(t) = \text{Re}[\tilde{\mathbf{m}}^{A(B)}e^{i\omega t}]$, where $\tilde{\mathbf{m}}^{A(B)} = \{\tilde{m}_x^{A(B)}e^{i\omega t}, \tilde{m}_y^{A(B)}e^{i\omega t}, 1\}$. By linearizing the LLG equations around the equilibrium position of each magnetic sublattice, we obtain

$$\tilde{m}_x^{A(B)} = \bar{\chi}_x^{A(B)}(\omega)\gamma\tilde{h}_{\text{rf}}, \quad (10a)$$

$$\tilde{m}_y^{A(B)} = \tilde{\chi}_y^{A(B)}(\omega)\gamma\tilde{h}_{\rm rf}. \tag{10b}$$

Therefore, the instantaneous power density $P_m(t)$ becomes

$$
\begin{aligned}
P_m(t) &= -\frac{\gamma m_s}{V_{\rm uc}}\,{\rm Re}[(\tilde{\chi}_x^A + \tilde{\chi}_x^B)i\omega e^{i\omega t}]\cos(\omega t)H_x^2 \\
&= -\frac{\gamma m_s}{V_{\rm uc}}|\tilde{\chi}_x^+|{\rm Re}[i\omega e^{i(\omega t + \phi^+)}]\cos(\omega t)H_x^2 \\
&= \frac{\gamma m_s}{V_{\rm uc}}|\tilde{\chi}_x^+|\omega\sin(\omega t + \phi^+)\cos(\omega t)H_x^2,
\end{aligned}
\tag{11}
$$

where we have defined $\tilde{\chi}_x^+ \equiv \tilde{\chi}_x^A + \tilde{\chi}_x^B = |\tilde{\chi}_x^+|e^{i\phi^+}$. Some straightforward algebra shows that

$$\tilde{\chi}_x^+(\omega) = (i\omega\alpha - \omega_A)\left[\frac{1}{D^+(\omega)} + \frac{1}{D^-(\omega)}\right], \tag{12}$$

where in the denominator,

$$D^\pm(\omega) \equiv (\omega \pm \omega_0)^2 - \omega_A(\omega_A + 2\omega_J) + 2i\omega(\omega_A + \omega_J)\alpha + (\omega\alpha)^2, \tag{13}$$

with $\omega_A = \gamma\mathcal{H}_\parallel$, $\omega_0 = \gamma\mathcal{H}_0$, and $\omega_J = \gamma\mathcal{H}_J$. By averaging over time, we find

$$\overline{P}_H = \overline{P}_m = \frac{m_s\gamma H_x^2}{2V_{\rm uc}}\omega|\tilde{\chi}_x^+(\omega)|\sin[\phi^+(\omega)], \tag{14}$$

which is maximized for $\phi^+ = \pi/2$ (at the resonance peak). Phenomenologically, $\phi^+ = \pi/2$ indicates that $\mathbf{h}_{\rm rf}(t)\cdot\mathbf{m}^{A(B)}(t) = 0$ so that $\mathbf{h}_{\rm rf}(t) \parallel \dot{\mathbf{m}}^{A(B)}(t)$.

Case II. When the driving field $\mathbf{h}_D$ is the NSOT field $\mathbf{h}_{\rm NS}$ generated by the electric component $\mathbf{E}_{\rm rf}$ of the electromagnetic wave, we have shown in ref. 17 in a generic context that

$$P_m = -m_s\alpha[(\dot{\mathbf{m}}^A)^2 + (\dot{\mathbf{m}}^B)^2]/\gamma V_{\rm uc}, \tag{15}$$

which, in a steady-state dynamics, balances the electrical power $P_E = \mathbf{J}^{\rm ad}\cdot\mathbf{E}$ with $\mathbf{J}^{\rm ad}$ being the adiabatic current density pumped by the magnetic dynamics, while the conduction current (i.e., Ohm's current) vanishes identically inside the insulator. For an $\mathbf{E}_{\rm rf} = E_x\cos(\omega t)\hat{\mathbf{x}}$ [Fig. 5b], the time-averaged electrical power density is

$$\overline{P}_E = \overline{P}_m = \frac{1}{2}E_x^2\,{\rm Re}[\tilde{\sigma}_{xx}(\omega)], \tag{16}$$

where $\tilde{\sigma}_{xx}(\omega)$ is the effective longitudinal conductivity, $\tilde{\sigma}_{xx} = \tilde{J}_x^{\rm ad}/\tilde{E}_x$. Microscopically, $\tilde{\sigma}_{xx}(\omega)$ is determined by the Berry curvature in the mixed momentum and magnetization space[12]:

$$\tilde{\sigma}_{xx} = i\omega\frac{\gamma e^2}{m_s V_{\rm uc}}\left\langle\Omega_{xx}^{km^A}\right\rangle^2\left[\tilde{\eta}_x^A(\omega) - \tilde{\eta}_x^B(\omega)\right], \tag{17}$$

where the Berry curvature $\Omega^{km}$ is the same as that in Eq. (3) and we have invoked the relation $\langle\Omega_{xx}^{km^A}\rangle = -\langle\Omega_{xx}^{km^B}\rangle$ for $\lambda_\nu = 0$. In this case, the dynamical susceptibility relates the NSOT field to the magnetic moments as $\tilde{m}_x^{A(B)} = \tilde{\eta}_x^{A(B)}(\omega)\gamma\tilde{h}_{\rm NS}^{A(B)}$ where

$$\tilde{h}_{\rm NS}^A = -\tilde{h}_{\rm NS}^B = -\frac{2\lambda_{ex}}{\hbar m_s}\delta\tilde{S}_x^A = -\frac{e}{m_s}\langle\Omega_{xx}^{km^A}\rangle\tilde{E}_x \tag{18}$$

is polarized along $x$ because $\mathbf{E}_{\rm rf}$ is polarized along $x$. Here we used a different notation $\tilde{\eta}$ to represent the dynamical susceptibility just to avoid confusion between Case II and Case I. We then arrive at the final

expression for the power absorption rate

$$
\begin{aligned}
\overline{P}_E &= \frac{\gamma e^2\omega E_x^2}{2m_s V_{\rm uc}}\left\langle\Omega_{xx}^{km^A}\right\rangle^2{\rm Re}[i(\tilde{\eta}_x^A - \tilde{\eta}_x^B)] \\
&= -\frac{\gamma e^2}{2m_s V_{\rm uc}}\left\langle\Omega_{xx}^{km^A}\right\rangle^2\omega|\tilde{\eta}_x^-(\omega)|\sin[\varphi^-(\omega)]E_x^2,
\end{aligned}
\tag{19}
$$

where we have defined $\tilde{\eta}_x^- \equiv \tilde{\eta}_x^A - \tilde{\eta}_x^B = |\tilde{\eta}_x^-|e^{i\varphi^-}$, for which we find

$$\tilde{\eta}_x^-(\omega) = (i\omega\alpha - \omega_A - 2\omega_J)\left[\frac{1}{D^+(\omega)} + \frac{1}{D^-(\omega)}\right], \tag{20}$$

with $D^\pm(\omega)$ defined in Eq. (13). Comparing with $\tilde{\chi}_x^+(\omega)$, an extra term $2\omega_J$ appears in the front factor of $\tilde{\eta}_x^-(\omega)$, which overwhelms $i\omega\alpha - \omega_A$ in absolute value. Similar to the previous case, $\overline{P}_E$ is maximized for $\varphi^- = \pi/2$ (at the resonance peak).

We can now compare the ratio of power absorption for the two cases:

$$\frac{\overline{P}_E}{\overline{P}_H} = \left[\frac{e\left\langle\Omega_{xx}^{km^A}\right\rangle}{m_s}\right]^2\left(\frac{E_x}{H_x}\right)^2\frac{|\tilde{\eta}_x^-|\sin\phi^-}{|\tilde{\chi}_x^+|\sin\phi^+} = \frac{h_{\rm NS}^2|\tilde{\eta}_x^-|\sin\varphi^-}{h_{\rm rf}^2|\tilde{\chi}_x^+|\sin\phi^+}, \tag{21}$$

where we have used $|\tilde{h}_{\rm rf}| = H_x$ and $|\tilde{h}_{\rm NS}| = |eE_x\langle\Omega_{xx}^{km^A}\rangle/m_s|$. As we set $h_{\rm NS} = h_{\rm rf}$ in the main text for comparing the two cases, this ratio can be evaluated at the resonance point where $\omega = \sqrt{\omega_A(\omega_A + 2\omega_E)} - \omega_0$ and $\varphi^-(\omega_r) \approx \phi^+(\omega_r) \approx \pi/2$:

$$\frac{\overline{P}_E}{\overline{P}_H} = \left\|\frac{i\omega_r\alpha - \omega_A - 2\omega_J}{i\omega_r\alpha - \omega_A}\right\| \approx 438.5. \tag{22}$$

A final remark: even in materials with much weaker SOC such that the NSOT field is, for example, an order of magnitude smaller than the magnetic field (i.e., $|\tilde{h}_{\rm NS}|/|\tilde{h}{\rm rf}| = 0.1$), the above ratio $\overline{P}_E/\overline{P}_H$ is 4.385— well above unity—indicating the robustness of the NSOT's enhanced efficiency in driving the AFM dynamics.

## Data availability

The data that support the findings of this study are all included or generated by the equations in the paper. All data are available from the corresponding authors upon reasonable request.

## Code availability

The source code of Kwant is available at https://kwant-project.org. The codes used in this study are available at the Github repository.

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

## Acknowledgements

The authors acknowledge fruitful discussions with E. del Barco, S. Singh, A. Kent and K. Deng. This work is supported by the National Science Foundation under Award No. DMR-2339315.

## Author contributions

R.C. conceived the project. J.T. performed the calculations with the help of H.Z., while all authors analyzed the data and discussed the results. J.T. and R.C. wrote the manuscript.

## Competing interests

The authors declare no competing interests.
