## [Peer Review File · Nature Communications]

Néel Spin-Orbit Torque in Antiferromagnetic Quantum Spin and Anomalous Hall Insulators

Corresponding Author: Dr Ran Cheng

Version 0:

Reviewer comments:

Reviewer #1

(Remarks to the Author)

In this manuscript Prof. Cheng and collaborators studied the antidamping-like Neel spin-orbit torque (NSOT) in a 2D honeycomb lattice model for antiferromagnetic (AFM) insulator and highlight the good efficiency of NSOT-driven AFM resonance. The idea is based on the valence band integrated Berry curvature nature of the SOT in insulators, and succeeds in applying it to AFM. Because NSOT in AFM insulators has not attracted considerable experimental attention, this study is meaningful in stimulating experimental researches in the new direction of AFM-insulator spintronics. Besides, although the valence band integrated Berry curvature nature of the SOT in insulators has been well known for spintronics scholars working with the Berry curvature theory, it seems not well realized by broader researchers of SOT community. In this regard, this work also has the potential to highlight the intriguing quantum geometric property of insulators to the whole community. Considering these exciting impacts, this work has the potential to be published in Nature Communications.

On the other hand, before a definite recommendation, I would like the authors to address the following questions and suggestions.

1. As mentioned, the valence band integrated Berry curvature nature of the SOT in insulators and the dissipationless driving of magnetic dynamics have been known before and been emphasized in several previous papers, such as PRB 98, 035123 (2018); PRB 104, 064433 (2021); and PRL 129, 086602 (2022). I recommend the authors to give credit to these works by citing them together with Refs. 12 and 13. Given that the theory of intrinsic SOT in insulators has been discussed, I recommend the authors to rephrase some claims on novelty. An example is "But all known scenarios are current-induced NSOT, which suffers from significant Joule heating.". Although experiment in AFM has not studied NSOT in insulators, the theoretical "scenario" of intrinsic Berry curvature is straightforward. A possible (just a suggestion) way to claim is "but previous experimental studies are limited to current-induced NSOT". Some other relevant claims in the manuscript may also be modified, such as the last sentence in the Abstract ".. unravel an incredible way to ..". These modifications will help convey more accurate information on the status of SOT research to general readers of Nature Communications. Meanwhile, I emphasize that these modifications do not affect the novelty and significance of this work as I mentioned in the first paragraph.

2. I feel that the discussion on topological phase is too long because it seems to have a little relation to NSOT. As one can see, Fig.2 and Fig.3 are both about topological phase and edge states. General readers may be confused by such an arrangement. In order to avoid such confusion, the authors have to add a sentence "the NSOT studied here is carried by the bulk adiabatic currents [12], which should not be confused with the edge currents discussed in the previous section." However, if this part is shorten or removed to supplemental Material, the information of the paper is more focused, and possible confusion can be avoided. If the authors would like to pertain the present arrangement, I think more clear connections between the topology and NSOT are necessary (which I think do not exist).

3. As insulator is considered, what is the origin of Γ in Eq. (2)? According to my understanding, Eq. (2) was derived for metallic systems by Zelezny et al., with a phenomenological relaxation rate Γ arising from electron scattering with disorder around the Fermi surface. In insulators without Fermi surface, the authors may wish to explain the origin of Γ .

4. In the last sentence of Abstract, the authors mentioned "Our findings .. to exploit AFM topological phases to achieve ultrafast magnetic dynamics." However, in the manuscript, it seems that only microwave driving is considered, which may

not belong to "ultrafast spintronics". I recommend the authors to reconsider the statement here. Besides, Eq. (2) is not applicable if the driving frequency is comparable to or close to the insulating gap, as shown explicitly in Ref. 14. I also recommend the authors to explicitly point this out in the manuscript.

Reviewer #3

(Remarks to the Author)

In this work titled "Néel Spin-Orbit Torque in Antiferromagnetic Quantum Spin and Anomalous Hall Insulators". The antiferromagnetic system contains two types of electrons. One type is the localized electron that contributes to local spin magnetic moments. The other one is valence electron that carry Berry curvatures in parameter spaces spanned by quasi-momentum and spin orientation of local spins. The presence of external electric field can modify the magnetization of topological valence electrons and thereby generate a spin torque on local spins. A dynamical electric field from microwave can generate a dynamical torque generating coherent spin precessions. The authors found that, compared to the torque from the magnetic field component of the microwave, the efficiency of the electric-field generated torque is much more efficient, promising low-energy consumption spintronics and magnonics.

This work is interesting. Nevertheless, the following concerns need to be addressed before I can recommend its publication in any form.

One concern is that spin torques in topological systems have been studied extensively by the authors and other groups. However, how this work is significantly distinguished from prior works of the authors and others is not clearly presented. Such difference is crucial for justifying its publication in Nature Communications.

Secondly, the efficiency of electric field driven torque is studied using one single set of parameter. I assume that the efficiency is closely related to the band gap. If the band gap is large enough, the effect of electric field will become smaller than the magnetic field in the microwave. This, however, is not clearly presented. For example, will the topological phase transition enhance the efficiency? Which kinds of transition can enhance the efficiency? Figure 4 shows some hints. But a comprehensive study will enhance the impact of this work.

Proposing realistic material candidates is also necessary. The authors mentioned MnPS₃. Do they use parameters such that the band gap is close to that of MnPS₃?

Minor:

- How is Eq. 2 derived?
- It would be helpful to identify the crucial symmetries related to each term in the Hamiltonian. For example, λ_v breaks $PT \otimes M_z$ symmetry of other terms making it possible to have nonzero Chern number. Positive and negative λ_R is related to each other by M_z operation, which does not change the Chern number, etc.
- How does the power absorption rate scale with Rashba SOC, sublattice potential, and λ_{ex} ?

Version 1:

Reviewer comments:

Reviewer #1

(Remarks to the Author)

I have read the detailed reply and the significantly revised manuscript. I appreciate very much the efforts of the authors in addressing both referees' comments and questions. I find my questions well answered by the authors. I recommend the publication of the present version.

Reviewer #3

(Remarks to the Author)

This is the second review report on the manuscript titled "Néel Spin-Orbit Torque in Antiferromagnetic Quantum Spin and Anomalous Hall Insulators". My main concerns regarding 1) the novelty compared to previous works and 2) the band gap dependence of the efficiency have been successfully addressed in the revised manuscript. Other minor points have also been addressed.

Thus, I would like to recommend its publication on Nature Communications in its current form.

Reviewer #1

“In this manuscript Prof. Cheng and collaborators studied the antidamping-like Neel spin-orbit torque (NSOT) in a 2D honeycomb lattice model for antiferromagnetic (AFM) insulator and highlight the good efficiency of NSOT-driven AFM resonance. The idea is based on the valence band integrated Berry curvature nature of the SOT in insulators and succeeds in applying it to AFM. Because NSOT in AFM insulators has not attracted considerable experimental attention, this study is meaningful in stimulating experimental research in the new direction of AFM-insulator spintronics. Besides, although the valence band integrated Berry curvature nature of the SOT in insulators has been well known for spintronics scholars working with the Berry curvature theory, it seems not well realized by broader researchers of SOT community. In this regard, this work also has the potential to highlight the intriguing quantum geometric property of insulators to the whole community. Considering these exciting impacts, this work has the potential to be published in Nature Communications. On the other hand, before a definite recommendation, I would like the authors to address the following questions and suggestions.”

Reply: We would like to thank the Reviewer for carefully reviewing our manuscript and providing informative feedback. We appreciate his/her overall positive assessment of our work. In the following, we append our point-by-point responses to all the comments and questions raised by the Reviewer.

1. *“As mentioned, the valence band integrated Berry curvature nature of the SOT in insulators and the dissipationless driving of magnetic dynamics have been known before and been emphasized in several previous papers, such as PRB 98, 035123 (2018); PRB 104, 064433 (2021); and PRL 129, 086602 (2022). I recommend the authors to give credit to these works by citing them together with Refs. 12 and 13. Given that the theory of intrinsic SOT in insulators has been discussed, I recommend the authors to rephrase some claims on novelty. An example is “But all known scenarios are current-induced NSOT, which suffers from significant Joule heating.”. Although experiment in AFM has not studied NSOT in insulators, the theoretical “scenario” of intrinsic Berry curvature is straightforward. A possible (just a suggestion) way to claim is “but previous experimental studies are limited to current-induced NSOT”. Some other relevant claims in the manuscript may also be modified, such as the last sentence in the Abstract “. unravel an incredible way to ..”. These modifications will help convey more accurate information on the status of SOT research to general readers of Nature Communications. Meanwhile, I emphasize that these modifications do not affect the novelty and significance of this work as I mentioned in the first paragraph.”*

Reply: We thank the Reviewer for this constructive comment. Our original intent was to emphasize that previous studies found either intrinsic SOT in topological insulators or

extrinsic NSOT driven by dissipative (Ohm's) currents, while intrinsic NSOT with combined merits of both categories have not been claimed so far.

We largely agree with the Reviewer that "*the theoretical 'scenario' of intrinsic Berry curvature is straightforward*", but we also would like to argue that when it comes to intrinsic NSOT, it is not as "straightforward" as the intrinsic SOT arising in topological systems. More importantly, the dynamical consequence of the intrinsic NSOT, especially its remarkably high efficiency in driving AFM resonance, remains elusive in existing studies.

To avoid over-claim, we have followed the Reviewer's suggestion and modified several statements in our manuscript to convey more accurate information and have cited all the papers mentioned by the Reviewer. All changes made have been highlighted in blue.

2. "*I feel that the discussion on topological phase is too long because it seems to have a little relation to NSOT. As one can see, Fig.2 and Fig.3 are both about topological phase and edge states. General readers may be confused by such an arrangement. In order to avoid such confusion, the authors have to add a sentence "the NSOT studied here is carried by the bulk adiabatic currents [12], which should not be confused with the edge currents discussed in the previous section."* However, if this part is shortened or removed to supplemental Material, the information of the paper is more focused, and possible confusion can be avoided. If the authors would like to pertain the present arrangement think more clear connections between the topology and NSOT are necessary (which I think do not exist)."

Reply: We thank the Reviewer for pointing out the potential distraction from the central story due to over-extended discussions on the topological phases. In the revised manuscript, we have shortened "Sec. III – Topological Phases" by merging Fig. 3(a) and (b) with Fig. 2, while leaving Fig. 3(c)-(f) (which are the real-space numerical plots of the edge currents) to the Supplemental Information. We also streamlined the discussions related to the new Fig. 2, making Sec. III a more concise story about topological phases.

We have also moved the sentence "*the NSOT studied here is carried by the bulk adiabatic currents, which should not be confused with the edge currents discussed in the previous section.*" To the Supplemental Materials.

In addition, to strengthen the subtle connection between band topology and the NSOT, we have added Fig. 4 (also suggested by Reviewer #2), which explicitly shows the variation of the non-equilibrium spin generation across different phases.

3. "*As insulator is considered, what is the origin of Γ in Eq. (2)? According to my understanding, Eq. (2) was derived for metallic systems by Zelezny et al., with a phenomenological relaxation rate Γ arising from electron scattering with disorder*

around the Fermi surface. In insulators without Fermi surface, the authors may wish to explain the origin of Γ .”

Reply: The Γ term comes from the self-energy in the Green’s function, which modifies both the Fermi-surface and the Fermi-sea contributions in the linear response theory. Therefore, both the extrinsic and intrinsic NSOT are subject to the band broadening Γ . The same technique had been adopted in several previous works, including Eq.(5) of [Zelezny *et al.*, PRL **113**, 157201] and Eq.(2) in the SM of [PNAS **117** (29) 16749]. In the revised manuscript, we have cited these papers when introducing Eq.(2). Note that these references called the Fermi-surface (Fermi-sea) contribution the “intra-band” (“inter-band”) contribution.

A critical yet subtle difference in the role of Γ between the extrinsic (Fermi-surface) and intrinsic (Fermi-sea) NSOT is that the former will diverge as $\Gamma \rightarrow 0$, while the latter remains finite when $\Gamma \rightarrow 0$, so long as we are not sitting on the phase boundaries.

In addition, with a finite broadening Γ , the intrinsic NSOT (among other physical quantities) will not diverge when crossing the phase boundaries where the band gap closes.

4. *“In the last sentence of Abstract, the authors mentioned “Our findings .. to exploit AFM topological phases to achieve ultrafast magnetic dynamics.” However, in the manuscript, it seems that only microwave driving is considered, which may not belong to “ultrafast spintronics”. I recommend the authors to reconsider the statement here. Besides, Eq. (2) is not applicable if the driving frequency is comparable to or close to the insulating gap, as shown explicitly in Ref. 14. I also recommend the authors to explicitly point this out in the manuscript.”*

Reply: Following the Reviewer’s suggestion, we have replaced “ultrafast” by “sub-terahertz” throughout the manuscript.

We have added a discussion in Sec. IV explaining that Eq.(2) becomes invalid when the driving frequency is comparable to the gap, which breaks the adiabatic condition that our formalism is based upon. We also mentioned that in the sub-terahertz frequency range, the adiabatic condition is very well respected, so Eq.(2) remains valid.

Reviewer #2

“In this work titled “Néel Spin-Orbit Torque in Antiferromagnetic Quantum Spin and Anomalous Hall Insulators”. The antiferromagnetic system contains two types of electrons. One type is the localized electron that contributes to local spin magnetic moments. The other one is valence electron that carry Berry curvatures in parameter spaces spanned by quasi-momentum and spin orientation of local spins. The presence of external electric field can modify the magnetization of topological valence electrons and thereby generate a spin torque on local spins. A dynamical electric field from microwave can generate a dynamical torque generating coherent spin precessions. The authors found that, compared to the torque from the magnetic field component of the microwave, the efficiency of the electric-field generated torque is much more efficient, promising low-energy consumption spintronics and magnonics. This work is interesting. Nevertheless, the following concerns need to be addressed before I can recommend its publication in any form.”

Reply: We are grateful to the Reviewer for carefully reviewing our manuscript and tendering a positive evaluation. Below please find our point-by-point responses.

1. *“One concern is that spin torques in topological systems have been studied extensively by the authors and other groups. However, how this work is significantly distinguished from prior works of the authors and others is not clearly presented. Such difference is crucial for justifying its publication in Nature Communications.”*

Reply: We agree with the Reviewer that spin torques in topological materials have been extensively studied by other groups and ourselves. However, this work presents novelty in two aspects:

- Previous studies were largely focused on the SOT, instead of NSOT, arising in topological materials, while our work is about NSOT in topological magnets.
- The NSOTs that have so far been claimed are limited to (Ohm’s) current-driven torques, which suffer from significant Joule heating. A celebrated example is the Mn_2Au , which we referenced as [15-20]. By contrast, our prediction is an *intrinsic* NSOT which in principle does not incur Joule heating because the conduction electrons are eliminated, while the NSOT is attributed to the *adiabatic* motions of topological electrons.

Concerning the distinctions between this work and previous works, please also check our reply to Q1 of Reviewer #1.

Another novelty of our work is that the same magnetic phase (defined by a collinear AFM order) supports both the QAH and the QSH phases without the aid of external magnetic fields, which, to the best of our knowledge, has not been claimed in previous studies.

In the revised manuscript, we have made the above points explicit. In particular, we have added the following statement in the Introduction.

“In stark contrast to the previously reported extrinsic NSOT driven by dissipative charge currents [25-30], our intrinsic NSOT is dissipationless and does not incur Ohm's conduction because only the adiabatic motions of valence electrons are involved. From the perspective of topological materials, our findings distinguish from previous studies by claiming the NSOT, rather than the widely-found SOT, in the topological nontrivial phases.”

2. *“Secondly, the efficiency of electric field driven torque is studied using one single set of parameters. I assume that the efficiency is closely related to the band gap. If the band gap is large enough, the effect of electric field will become smaller than the magnetic field in the microwave. This, however, is not clearly presented. For example, will the topological phase transition enhance the efficiency? Which kinds of transition can enhance the efficiency? Figure 4 shows some hints. But a comprehensive study will enhance the impact of this work.”*

Reply: We thank the Reviewer for pointing out this important issue.

To answer the first half of this question concerning the efficiency related to the band gap, we have promoted a critical relation from an in-line equation to Eq.(5), basing on which we tendered a discussion on how the effective NSOT field depends on the exchange coupling λ_{ex} . While $\delta N = (\delta S^A - \delta S^B)/2$ decreases monotonically with an increasing λ_{ex} according to Eq.(2), where the band gap in the integrand is approximately proportional to λ_{ex} (when $\lambda_{ex} \gg \lambda_R, \lambda_{soc}, \lambda_v$), there is an additional λ_{ex} in Eq.(5) which largely compensates the λ_{ex} -dependence of δN , rendering the overall efficiency of driving quite insensitive to λ_{ex} .

To address the second half of this question concerning the influence of phase transitions on the efficiency, we have added a new figure followed by extended discussions. Below is the Fig. 4 in the revised manuscript (while we have merged former Fig. 3 with Fig. 2).

We also made a final remark at the end of *Method*: “*even in materials with much weaker SOC such that the NSOT field is, for example, an order of magnitude smaller than the magnetic field (i.e., $|\tilde{h}_{NS}|/|\tilde{h}_{rf}| = 0.1$), the above ratio \bar{P}_E/\bar{P}_H is 4.385—well above unity—indicating the robustness of the NSOT’s enhanced efficiency in driving the AFM dynamics.*”

3. “*Proposing realistic material candidates is also necessary. The authors mentioned MnPS₃. Do they use parameters such that the band gap is close to that of MnPS₃?*”

Reply: Our work is a general theoretical proposal of a new mechanism and its ensuing phenomena, which is not supposed to be restricted to a particular material. We mentioned TMTs (such as MnPS₃ and its variances) just to motivate the discussion, while our findings can be applied to a wider class of materials describable by our Hamiltonian. Therefore, without compromising the generality of our theory, we have used empirical parameters in TMTs, as we don’t want to strictly stick to their exact values in MnPS₃. We made necessary modifications on our statements in the revised manuscript.

As for the case of MnPS₃, the band gap is about 2 eV according to recent DFT studies. In our general model, the topological band gap is roughly $2\lambda_{ex}$ (assuming $\lambda_{ex} \gg \lambda_R, \lambda_{SOC}, \lambda_v$) and we have used $\lambda_{ex} = 0.1t = 100$ meV. To reproduce a 2 eV gap, we could just amplify the universal scaling factor t by 10 times, which will not modify any phase diagrams where all parameters have been scaled by the same t . Indeed, such a rescaling will not change the efficiency of NSOT at all.

If we only enlarge λ_{ex} while keeping everything else the same, however, we could as well easily reproduce the 2 eV gap but at the cost of significant changes in the phase diagrams. Nevertheless, inside the same phase, it is likely that the efficiency of NSOT will not change too much per our argument in addressing Q2.

4. “*How is Eq. 2 derived?*”

Reply: Eq.(2) can be straightforwardly obtained by using the Kubo formula (linear response theory). Specifically, $\delta\mathbf{N} = (\delta\mathbf{S}^A - \delta\mathbf{S}^B)/2$, where $\delta S_i^{A/B} = \chi_{ij}^{(A/B)} E_j$ (i, j are the real-space components) for each sublattice. Similar formulas can be found in [PRL **113**, 157201] and [PRB **80**, 134403], which we have cited in the revised manuscript. The Pauli matrix τ_3 in Eq.(2) takes care of the minus sign in the definition $\delta\mathbf{N} = (\delta\mathbf{S}^A - \delta\mathbf{S}^B)/2$. In the clean limit $\Gamma \rightarrow 0$, Eq.(2) will reduce to the Berry curvature residing in the mixed space spanned by the momentum \mathbf{k} and the magnetization \mathbf{m} , which had been discussed extensively in our recent works [PRB **106**, 054418] and [PRL **132**, 136701].

In this regard, with proper citations of these previous works, it is not necessary to lay out a detailed derivation of Eq.(2).

5. “It would be helpful to identify the crucial symmetries related to each term in the Hamiltonian. For example, λ_v breaks PT symmetry of other terms making it possible to have nonzero Chern number. Positive and negative λ_R is related to each other by M_z operation, which does not change the Chern number, etc.”

Reply: We thank the Reviewer for this helpful suggestion. In the revised manuscript, we referred to symmetry analysis at several places in Sec. II and III.

6. “How does the power absorption rate scale with Rashba SOC, sublattice potential, and λ_{ex} ?”

Reply: Per Eq. (3) and Eq. (18), we can eliminate $\langle \omega_{xx}^{km} \rangle^2$ and arrive at an overall power absorption rate $P_E \propto \lambda_{ex}^2 [(\delta S^A)^2 + (\delta S^B)^2] \propto \lambda_{ex}^2 (\delta N)^2$, where the dependences of $\delta S^{A,B}$ (hence δN) on λ_{soc} and λ_R are shown by the phase diagrams Fig. 3 and Fig. 4.

As mentioned in our reply to Q2 above, the topological band gap is roughly proportional to λ_{ex} (when $\lambda_{ex} \gg \lambda_R, \lambda_{soc}, \lambda_v$), so δN calculated by Eq.(2) is (approximately) inversely proportional to λ_{ex} , leaving $P_E \propto \lambda_{ex}^2 (\delta N)^2$ insensitive to λ_{ex} . This fact is consistent with the insensitivity of the NSOT strength to λ_{ex} .

List of all significant changes

- Section I has been enriched with numerous clarifications.
- Section II has been shortened, reorganized, and streamlined.
- Former Figs.3(a) & (b) have been merged into the new Fig.2, while former Figs.3 (c)-(f) have been moved to the SI.
- A new figure (Fig.4) has been added, followed by extended discussions.
- Equation (5) is added, followed by extended discussions.
- The results in the new Fig.3 (former Fig.4) have been updated with a denser mesh.
- References [13-16], [26] and [41] have been added.
- Fig. S3 in the SI has been replaced with a new version.
- Discussions and clarifications requested by the reviewers have been added from place to place.